# Adversarial Neuron Pruning Purifies Backdoored Deep Models

**Dongxian Wu**[1]*     **Yisen Wang**[2,3]†
[1]Dept. of Computer Science and Technology, Tsinghua University, China
[2]Key Lab. of Machine Perception, School of Artificial Intelligence, Peking Univesity, China
[3]Institute for Artificial Intelligence, Peking Univesity, China

## Abstract

As deep neural networks (DNNs) are growing larger, their requirements for computational resources become huge, which makes outsourcing training more popular. Training in a third-party platform, however, may introduce potential risks that a malicious trainer will return backdoored DNNs, which behave normally on clean samples but output targeted misclassifications whenever a trigger appears at the test time. Without any knowledge of the trigger, it is difficult to distinguish or recover benign DNNs from backdoored ones. In this paper, we first identify an unexpected sensitivity of backdoored DNNs, that is, they are much easier to collapse and tend to predict the target label on clean samples when their neurons are adversarially perturbed. Based on these observations, we propose a novel model repairing method, termed *Adversarial Neuron Pruning* (ANP), which prunes some sensitive neurons to purify the injected backdoor. Experiments show, even with only an extremely small amount of clean data (*e.g.*, 1%), ANP effectively removes the injected backdoor without causing obvious performance degradation. Our code is available at `https://github.com/csdongxian/ANP_backdoor`.

## 1 Introduction

In recent years, deep neural networks (DNNs) achieve satisfactory performance in many tasks, including computer vision [15], speech recognition [45], and gaming agents [36]. However, their success heavily relies on a large amount of computation and data, forcing researchers to outsource the training in "Machine Learning as a Service" (MLaaS) platforms or download pretrained models from the Internet, which brings potential risks of training-time attacks [17, 35, 2]. Among them, backdoor attack [14, 6, 42] is remarkably dangerous because it stealthily builds a strong relationship between a trigger pattern and a target label inside DNNs by poisoning a small proportion of the training data. As a result, the returned model always behaves normally on the clean data but is controlled to make target misclassification by presenting the trigger pattern such as a specific single-pixel [41] or a black-white checkerboard [14] at the test time.

Since DNNs are deployed in many real-world and safety-critical applications, it is urgent to defend against backdoor attacks. While there are many defense methods during training [38, 41, 8, 29, 20], this work focuses on a more realistic scenario in outsourcing training that repairs models after training. In particular, the defenders try to remove the injected backdoor without access to the model training process. Without knowledge of the trigger pattern, previous methods only achieve limited robustness [7, 24, 22]. Some works try to reconstruct the trigger [43, 5, 33, 50, 44], however, the

---

*Work was done as an internship at Peking University when he was a student at Tsinghua University. Now, he is a Post-doc at the University of Tokyo.

†Corresponding author: Yisen Wang (yisen.wang@pku.edu.cn).

trigger pattern in brand-new attacks becomes natural [26], invisible [49], and dynamic [30], leading to the reconstruction infeasible. Different from them, in this paper, we turn to explore *whether we can successfully remove the injected backdoor even without knowing the trigger pattern*.

Usually, the adversary perturbs inputs to cause misclassification, *e.g.*, attaching triggers for backdoored models or adding adversarial perturbations. Parallel to this, we are also able to cause misclassification by perturbing neurons of DNNs [47]. For any neuron inside DNN, we could perturb its weight and bias by multiplying a relatively small number, to change its output. Similar to adversarial perturbations, we can optimize the neuron perturbations to increase its classification loss. Surprisingly, we find that, within the same perturbation budget, backdoored DNNs are much easier to collapse and prone to output the target label than normal DNNs even without the presence of the trigger. That is, the neurons that are sensitive to adversarial neuron perturbation are strongly related to the injected backdoor. Motivated by this, we propose a novel model repairing method, named *Adversarial Neuron Pruning* (ANP), which prunes most sensitive neurons under the adversarial neuron perturbation. Since the number of neurons is much smaller than weight parameters, *e.g.*, only 4810 neurons for ResNet-18 while 11M parameters for the same model, our method can work well only based on 1% of clean data. Our main contributions are summarized as follows:

- We find that through adversarially perturbing neurons, backdoored DNNs can present backdoor behaviors even without the presence of the trigger patterns and are much easier to output misclassification than normal DNNs.

- To defend backdoor attacks, we propose a simple yet effective model repairing method, *Adversarial Neuron Pruning* (ANP), which prunes the most sensitive neurons under adversarial neuron perturbations without fine-tuning.

- Extensive experiments demonstrate that ANP consistently provides state-of-the-art defense performance against various backdoor attacks, even using an extremely small amount of clean data.

## 2 Related Work

### 2.1 Backdoor Attacks

The backdoor attack is a type of attacks occurring during DNN training. The adversary usually poisons a fraction of training data via attaching a predefined trigger and relabeling them as target labels (dirty-label setting) [14, 6]. All poisoned samples can be relabeled as a single target class (all-to-one), or poisoned samples from different source classes are relabeled as different target classes (all-to-all) [30]. After training, the model can be controlled to predict the target label in the presence of the trigger at the test time. Different from evasion attacks (*e.g.*, adversarial attacks) [3, 39, 13], backdoor attacks aim to embed a input- and model-agnostic trigger into the model, which is a severe threat to the applications of deep learning [12]. Since incorrectly-labeled samples are easy to be detected, some attacks attach the trigger to samples from the target class (clean-label setting) [35, 42, 1]. Apart from simple forms like a single-pixel [41] or a black-and-white checkerboard [14], the trigger patterns can be more complex such as a sinusoidal strip [1] and a dynamic pattern [30]. These triggers in recent attacks become more natural [26] and human-imperceptible [49, 31], making them stealthy and hard to be detected by human inspection. Besides, powerful adversaries who have access to the model can optimize the trigger pattern [25], and even co-optimize the trigger pattern and the model together to enhance the power of backdoor attacks [32].

### 2.2 Backdoor Defense

**Defense during Training.** With access to training, the defender is able to detect the poisoned data or make poisoning invalid during the training process. Regarding poisoned data as outliers, we can detect and filter them out using robust statistics in the input space [38, 18, 8, 9] or the feature space [17, 41, 28, 10]. Meanwhile, other studies focus on training strategies to make the poisoned data have little or no effect on the trained model [23, 40] through randomized smoothing [34, 46], majority vote [21], differential privacy [29], and input preprocessing [27, 4].

**Defense after Training.** For a downloaded model, we have lost control of its training. To repair the risk model, one direct way is to first reconstruct an approximation of the backdoor trigger via

adversarial perturbation [43] or generative adversarial network (GAN) [5, 33, 50]. Once the trigger is reconstructed, it is feasible to prune neurons that are activated in the presence of the trigger, or fine-tune the model to unlearn the trigger [43]. As the trigger patterns in recently proposed attacks become more complicated such as the dynamics trigger [30] or natural phenomenon-based trigger [26], reconstruction becomes increasingly difficult. There are other studies on trigger-agnostic repairing via model pruning [24] or fine-tuning [7, 22] on the clean data. While they may suffer from severe accuracy degradation when only small clean data are available [7]. Unlike previous defensive pruning methods which are based on the rule-of-thumb, our proposed method is data-driven and does not require additional fine-tuning.

## 3 The Proposed Method

### 3.1 Preliminary

**Deep Neural Network and Its Neurons.** In this paper, we take a fully-connected network as an example (other convolutional networks are also applicable). Its layers are numbered from 0 (input) to $L$ (output) and each layer contains $n_0, \cdots, n_L$ neurons. The network has $n = n_1 + n_2 + \cdots + n_L$ parameterized neurons in total[3]. We denote the weight parameters of the $k$-th neuron in $l$-th layer as $\mathbf{w}_k^{(l)}$, the bias as $b_k^{(l)}$, and its output is

$$h_k^{(l)} = \sigma(\mathbf{w}_k^{(l)\top} \mathbf{h}^{(l-1)} + b_k^{(l)}), \tag{1}$$

where $\sigma(\cdot)$ is the nonlinear function and $\mathbf{h}^{(l-1)}$ is the outputs of all neurons in the previous layer, *i.e.*, $\mathbf{h}^{(l-1)} = [h_1^{(l-1)}, \cdots, h_{n_l}^{(l-1)}]$. For simplicity, we denote all weights of the network as $\mathbf{w} = [\mathbf{w}_1^{(1)}, \cdots, \mathbf{w}_{n_1}^{(1)}, \cdots, \mathbf{w}_1^{(L)}, \cdots, \mathbf{w}_{n_L}^{(L)}]$, and all biases as $\mathbf{b} = [b_1^{(1)}, \cdots, b_{n_l}^{(1)}, \cdots, b_1^{(L)}, \cdots, b_{n_L}^{(L)}]$. For a given input $\mathbf{x}$, the network makes the prediction $f(\mathbf{x}; \mathbf{w}, \mathbf{b})$.

**DNN Training.** We consider a $c$-class classification problem. The parameters of weights and biases in DNN are learned on a training dataset $\mathcal{D}_\mathcal{T} = \{(\mathbf{x}_1, y_1), \cdots, (\mathbf{x}_s, y_s)\}$ containing $s$ inputs, where each input is $\mathbf{x}_i \in \mathbb{R}^d, i = 1, \cdots, s$, and its ground-truth label is $y_i \in \{1, \cdots, c\}$. The training procedure tries to find the optimal $(\mathbf{w}^\star, \mathbf{b}^\star)$ which minimize the training loss on the training data $\mathcal{D}_\mathcal{T}$,

$$\mathcal{L}_{\mathcal{D}_T}(\mathbf{w}, \mathbf{b}) = \mathop{\mathbb{E}}_{\mathbf{x}, y \sim \mathcal{D}_T} \ell(f(\mathbf{x}; \mathbf{w}, \mathbf{b}), y), \tag{2}$$

where $\ell(\cdot, \cdot)$ is usually cross entropy loss.

**Defense Setting.** We adopt a typical defense setting that an untrustworthy model is downloaded from a third party (*e.g.*, outsourcing training) without knowledge of the training data $\mathcal{D}_\mathcal{T}$. For defense, we are assumed to have a small amount of clean data $\mathcal{D}_\mathcal{V}$. The goals of model repairing are to remove the backdoor behavior while keeping the accuracy on clean samples.

### 3.2 Adversarial Neuron Perturbations

In previous studies, the adversary causes misclassification by perturbing inputs before feeding them to DNNs. For example, attaching triggers to inputs to make backdoored DNN output the target label or adversarially perturbing inputs to make normal/backdoored DNN output incorrectly.

Here, we try another direction to perturb DNN neurons to cause misclassification. Given the $k$-th neuron in the $l$-th layers, we can perturb its weight $\mathbf{w}_k^{(l)}$ and bias $b_k^{(l)}$ by multiplying a small number respectively. As a result, the perturbed weight is $(1 + \delta_k^{(l)})\mathbf{w}_k^{(l)}$, the perturbed bias is $(1 + \xi_k^{(l)})b_k^{(l)}$, and the output of the perturbed neuron becomes

$$h_k^{(l)} = \sigma\big((1 + \delta_k^{(l)})\mathbf{w}_k^{(l)\top}\mathbf{h}^{(l-1)} + (1 + \xi_k^{(l)})b_k^{(l)}\big), \tag{3}$$

where $\delta_k^{(l)}$ and $\xi_k^{(l)}$ indicate the relative sizes of the perturbations to the weight and bias respectively. Similarly, all neurons can be perturbed like Eq. (3). We denote the relative weight perturbation size of all neurons as $\boldsymbol{\delta} = [\delta_1^{(1)}, \cdots, \delta_{n_1}^{(1)}, \cdots, \delta_1^{(L)}, \cdots, \delta_{n_L}^{(L)}]$, and the relative bias perturbation

---

[3]The input neurons are not considered since they have no parameters.

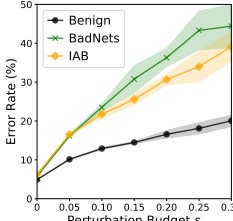 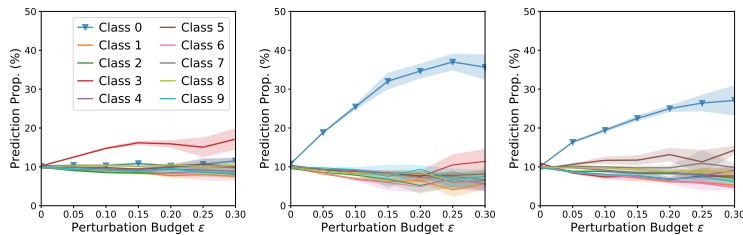

(a) The error rate ($\pm$ std) of different models under neuron perturbations.

(b) The proportion ($\pm$ std over 5 random runs) of different classes in predictions by a benign model (*Left*) and two models backdoored by BadNets (*Middle*) and IAB attack (*Right*) under neuron perturbations.

Figure 1: The performance of neuron-perturbed models with different perturbation budgets.

size as $\boldsymbol{\xi} = [\xi_1^{(1)}, \cdots, \xi_{n_1}^{(1)}, \cdots, \xi_1^{(L)}, \cdots, \xi_{n_L}^{(L)}]$ for simplicity. For a clean input $\mathbf{x}$, the output of the perturbed model is

$$f(\mathbf{x}; (1 + \boldsymbol{\delta}) \odot \mathbf{w}, (1 + \boldsymbol{\xi}) \odot \mathbf{b}), \tag{4}$$

where neuron-wise multiplication $\odot$ multiplies the parameters by the perturbation sizes belonging to the same neuron as follows:

$$(1 + \boldsymbol{\delta}) \odot \mathbf{w} = \left[(1 + \delta_1^{(1)})\mathbf{w}_1^{(1)}, \cdots, (1 + \delta_{n_1}^{(1)})\mathbf{w}_{n_1}^{(1)}, \cdots, (1 + \delta_1^{(L)})\mathbf{w}_1^{(L)}, \cdots, (1 + \delta_{n_L}^{(L)})\mathbf{w}_{n_L}^{(L)}\right], \tag{5}$$

$$(1 + \boldsymbol{\xi}) \odot \mathbf{b} = \left[(1 + \xi_1^{(1)})b_1^{(1)}, \cdots, (1 + \xi_{n_1}^{(1)})b_{n_1}^{(1)}, \cdots, (1 + \xi_1^{(L)})b_1^{(L)}, \cdots, (1 + \xi_{n_L}^{(L)})b_{n_L}^{(L)}\right]. \tag{6}$$

Given a trained model, similar to adversarial perturbations [39, 13], we optimize the neuron perturbations $\boldsymbol{\delta}$ and $\boldsymbol{\xi}$ to increase the classification loss on the clean data:

$$\max_{\boldsymbol{\delta}, \boldsymbol{\xi} \in [-\epsilon, \epsilon]^n} \mathcal{L}_{\mathcal{D}_\mathcal{V}}((1 + \boldsymbol{\delta}) \odot \mathbf{w}, (1 + \boldsymbol{\xi}) \odot \mathbf{b}), \tag{7}$$

where $\epsilon$ is the perturbation budget, which limits the maximum perturbation size.

Specifically, we generate adversarial neuron perturbations for three ResNet-18 models [15] on CIFAR-10 [19]: a benign model, a backdoored one with a predefined trigger (Badnets [14]), and a backdoored one with a dynamic trigger (Input-aware Backdoor Attack [30], IAB). We all set the target label in backdoor as class 0. We solve the maximization in Eq. (7) using project gradient descent (PGD) with random initialization similar to generating adversarial examples. The number of iterations is 300 and the step size is 0.001. Figure 1(a) shows adversarial neuron perturbations leads to misclassification, and the larger the perturbation budget $\epsilon$ is, the larger the error rate is. In addition, backdoored models always have larger error rates with the same perturbation budget compared to the benign one. To explore the misclassification in detail, we illustrate the proportion of different classes in predictions by three models in Figure 1(b). For benign model, adversarial neuron perturbations mislead the model to output the untargeted class (*e.g.*, class 3). While for backdoored models, the majority of misclassified samples are predicted as the target label (*e.g.*, class 0) whose proportion rate is much higher than the benign model. More results to support these findings can be found in Appendix B. Therefore, even without the trigger, we can still induce the backdoor behaviours via adversarial neuron perturbations. The reasons can be explained as follows: assuming the $k$-th neuron in $l$-th layers is backdoor-related, it is dormant on clean sample [24] (*i.e.*, output is close to 0: $h_k^{(l)} \approx 0$). If $\mathbf{w}_k^{(l)\top}\mathbf{h}^{l-1} > 0$ and $b_k^{(l)} > 0$, we can increase its output by enlarging the norm of the weights (*i.e.*, $\delta_k^{(l)} > 0$), and further continue to increase it by adding a larger bias (*i.e.*, $\xi_k^{(l)} > 0$)[4]. With suitable perturbations, the backdoor-related neuron is activated even on clean samples. Thus, the neurons that are sensitive to adversarial neuron perturbation are strongly related to the injected backdoor.

### 3.3 The Proposed Adversarial Neuron Pruning (ANP)

As mentioned above, the sensitivity under adversarial neuron perturbations is strongly related to the injected backdoor. Inspired by this, we try to prune some sensitive neurons to defend backdoor

---

[4]If $\mathbf{w}_k^{(l)\top}\mathbf{h}^{l-1} < 0$ or $b_k^{(l)} < 0$, we still can increase its output by reducing the norm of the weight or bias.

attacks, named Adversarial Neuron Pruning method (ANP). We denote the pruning masks as $\mathbf{m} = [m_1^{(1)}, \cdots, m_{n_1}^{(1)}, \cdots, m_1^{(L)}, \cdots, m_{n_L}^{(L)}] \in \{0, 1\}^n$. For the $k$-th neuron in the $l$-th layer, we set its weight $\mathbf{w}_k^{(l)} = \mathbf{0}$ if $m_k^{(l)} = 0$, and keep it unchanged if $m_k^{(l)} = 1$. Similar to Ghorbani and Zou [11], we always keep the bias $b_k^{(l)}$ to avoid extra harm to the accuracy on clean data[5].

**Continuous Relaxation and Optimization.** Due to the binary masks, pruning is a discrete optimization problem that is difficult to solve within feasible time. To address this, we apply continuous relaxation to let $\mathbf{m} \in [0, 1]^n$ and optimizes them using projected gradient descent (PGD). To restore the continuous masks to discreteness after optimization, we set all mask smaller than a threshold as 0 (prune neurons with small masks), and others as 1 (keep neurons with large masks). We expect the pruned model to not only behaves well on clean data but also keep stable under adversarial neuron perturbations. Therefore, we solve the following minimization problem,

$$\min_{\mathbf{m} \in [0,1]^n} \left[ \alpha \mathcal{L}_{\mathcal{D}_{\mathcal{V}}}(\mathbf{m} \odot \mathbf{w}, \mathbf{b}) + (1 - \alpha) \max_{\boldsymbol{\delta}, \boldsymbol{\xi} \in [-\epsilon, \epsilon]^n} \mathcal{L}_{\mathcal{D}_{\mathcal{V}}}((\mathbf{m} + \boldsymbol{\delta}) \odot \mathbf{w}, (1 + \boldsymbol{\xi}) \odot \mathbf{b}) \right], \quad (8)$$

where $\alpha \in [0, 1]$ is a trade-off coefficient. If $\alpha$ is close to 1, the minimization object focuses more on accuracy of the pruned model on clean data, while if $\alpha$ is close to 0, it focuses more on robustness against backdoor attacks.

**Implementation of ANP.** As shown in Algorithm 1, we start from an unpruned network, that is, initializing all masks as 1 (Line 2). We randomly initialize perturbations (Line 5), and update them using one step (Line 6-7), then project them onto the feasible regime (Line 8-9). Following that, we update mask values based on the adversarial neuron perturbation (Line 10) and project mask value onto $[0, 1]$ (Line 11). As convergence after iterations, we pruned neurons with small mask value and keep others unchanged (Line 13). Note that the proposed ANP is data-driven, different from previous pruning-based defenses which are based on the thumb-of-rule [24, 43]. Besides, ANP can work well on an extremely small amount of clean data due to the small number of masks.

---

**Algorithm 1** Adversarial Neuron Pruning (ANP)

1: **Input:** Network $f(\cdot; \mathbf{w}, \mathbf{b})$, hyper-parameter $\alpha$, learning rate $\eta$, batch size $b$, maximum perturbation size $\epsilon$
2: Initialize all elements in $\mathbf{m}$ as 1
3: **repeat**
4:      Read mini-batch $\mathcal{B} = \{(\mathbf{x}_1, y_1), ..., (\mathbf{x}_b, y_b)\}$ from training set
5:      $\boldsymbol{\delta}_0, \boldsymbol{\xi}_0 \sim U(-\epsilon, \epsilon)$, where $U(-\epsilon, \epsilon)$ is the uniform distribution
6:      $\boldsymbol{\delta} \leftarrow \boldsymbol{\delta}_0 + \epsilon \text{sign}(\nabla_{\boldsymbol{\delta}} \mathcal{L}((\mathbf{m} + \boldsymbol{\delta}_0) \odot \mathbf{w}, (1 + \boldsymbol{\xi}_0) \odot \mathbf{b}))$
7:      $\boldsymbol{\xi} \leftarrow \boldsymbol{\xi}_0 + \epsilon \text{sign}(\nabla_{\boldsymbol{\xi}} \mathcal{L}((\mathbf{m} + \boldsymbol{\delta}_0) \odot \mathbf{w}, (1 + \boldsymbol{\xi}_0) \odot \mathbf{b}))$
8:      $\boldsymbol{\delta} \leftarrow \max(-\epsilon, \min(\boldsymbol{\delta}, \epsilon))$
9:      $\boldsymbol{\xi} \leftarrow \max(-\epsilon, \min(\boldsymbol{\xi}, \epsilon))$
10:      $\mathbf{m} \leftarrow \mathbf{m} - \eta \nabla_{\mathbf{m}} [\alpha \mathcal{L}(\mathbf{m} \odot \mathbf{w}, \mathbf{b}) + (1 - \alpha) \mathcal{L}((\mathbf{m} + \boldsymbol{\delta}) \odot \mathbf{w}, (1 + \boldsymbol{\xi}) \odot \mathbf{b})]$
11:      $\mathbf{m} \leftarrow \max(0, \min(\mathbf{m}, 1))$
12: **until** training converged
13: $m_k^{(l)} = \mathbb{I}(m_k^{(l)} > threshold), \text{for all } k, l$
14: **Output:** A robust network $f(\cdot; \mathbf{m} \odot \mathbf{w}, \mathbf{b})$ against backdoor attacks

---

**Adaptation to Batch Normalization.** Batch Normalization (BatchNorm) [16] always normalizes its input and controls the mean and variance of the output by a pair of trainable parameters (scale $\gamma$ and shift $\beta$). If BatchNorm is inserted between matrix multiplication and the nonlinear activation, the perturbations to weight and bias may cancel each other out. For example, if we perturb the weight and bias of a neuron to the maximum by multiplying $(1 + \epsilon)$, the normalization offsets them and nothing changes after BatchNorm. To address this problem, we perturb the scale and shift parameters instead. We also make similar changes to the pruning masks.

---

[5]We can still compress the model since these biases can be absorbed by their following layers.

Table 1: Performance (average over 5 random runs) of 4 defense methods against 6 backdoor attacks on $1\%$ (500 images) of clean data on CIFAR-10 training set using ResNet-18. The *AvgDrop* indicates the average changes in ACC or ASR over 6 backdoor attacks compared to no defense results (*Before*).

| Metric | Defense | Badnets | Blend | IAB-one | IAB-all | CLB | SIG | AvgDrop |
|--------|---------|---------|-------|---------|---------|-----|-----|---------|
| ACC | Before | 93.73 | 94.82 | 93.89 | 94.10 | 93.78 | 93.64 | – |
|  | FT($lr = 0.01$) | 90.48 | 92.12 | 88.68 | 89.06 | 91.26 | 91.19 | $\downarrow 3.53$ |
|  | FT($lr = 0.02$) | 87.23 | 88.98 | 84.85 | 83.77 | 88.25 | 88.63 | $\downarrow 7.04$ |
|  | FP | **92.18** | 92.40 | 91.57 | 92.28 | 91.91 | 91.64 | $\downarrow 2.00$ |
|  | MCR($t = 0.3$) | 85.95 | 88.26 | 86.30 | 84.53 | 86.87 | 85.88 | $\downarrow 7.70$ |
|  | ANP | 90.20 | **93.44** | **92.62** | **92.79** | **92.67** | **93.40** | $\downarrow$**1.47** |
| ASR | Before | 99.97 | 100.0 | 98.49 | 92.88 | 99.94 | 94.26 | – |
|  | FT($lr = 0.01$) | 11.70 | 47.17 | 0.99 | 1.36 | 12.51 | 0.40 | $\downarrow 85.24$ |
|  | FT($lr = 0.02$) | 2.95 | 10.20 | 1.70 | 1.83 | **1.17** | 0.39 | $\downarrow 94.55$ |
|  | FP | 5.34 | 65.39 | 20.73 | 32.36 | 3.40 | 0.32 | $\downarrow 76.33$ |
|  | MCR($t = 0.3$) | 5.70 | 13.57 | 30.23 | 35.17 | 12.77 | 0.52 | $\downarrow 81.26$ |
|  | ANP | **0.45** | **0.46** | **0.88** | **0.86** | 3.98 | **0.28** | $\downarrow$**96.44** |

## 4 Experiments

In this section, we conduct comprehensive experiments to evaluate the effectiveness of ANP, including its benchmarking robustness, ablation studies and performance under limited computation resources.

### 4.1 Experimental Settings

**Backdoor Attacks and Settings.** We consider 5 state-of-the-art backdoor attacks: 1) BadNets [14], 2) Blend backdoor attack (Blend) [6], 3) Input-aware backdoor attack with all-to-one target label (IAB-one) or all-to-all target label (IAB-all) [30], 4) Clean-label backdoor (CLB) [42], and 5) Sinusoidal signal backdoor attack (SIG) [1]. For fair comparisons, we follow the default configuration in their original papers such as the trigger patterns, the trigger sizes and the target labels. We evaluate the performance of all attacks and defenses on CIFAR-10 [19] using ResNet-18 [15] as the base model. We use $90\%$ training data to train the backdoored DNNs and use the all or part of the remaining $10\%$ training data for defense. More details about implementation can be found in Appendix A.

**Backdoor Defenses and Settings.** We compare our proposed ANP with 3 existing model repairing methods: 1) standard fine-tuning (FT), 2) fine-pruning (FP) [24], and 3) mode connectivity repair (MCR) [48]. All defense methods are assumed to have access to the same $1\%$ of clean training data (500 images). The results based on $10\%$ (5000 images) and $0.1\%$ (50 images) of clean training data can be found in Appendix. For ANP, we optimize all masks using Stochastic Gradient Descent (SGD) with the perturbation budget $\epsilon = 0.4$ and the trade-off coefficient $\alpha = 0.2$. We set the batch size 128, the constant learning rate 0.2, and the momentum 0.9 for 2000 iterations in total. Typical data augmentation like random crop and horizontal flipping are applied. After optimization, neurons with mask value smaller than 0.2 are pruned.

**Evaluation Metrics.** We evaluate the performance of different defenses using two metrics: 1) the accuracy on clean data (ACC), and 2) and the attack success rate (ASR) that is the ratio of triggered samples that are misclassified as the target label. For better comparison between different strategies for the target label (*e.g.*, all-to-one and all-to-all), we remove the samples whose ground-truth labels already belong to the target class in the all-to-one setting before calculating ASR. Therefore, an ideal defense always has close-to-zero ASR and high ACC.

### 4.2 Benchmarking the State-of-the-art Robustness

To verify the effectiveness of the proposed ANP, we compare its performance with other 3 existing model repairing methods using ACC and ASR in Table 1. All experiments are repeated over 5 runs with different random seeds, and we only report the average performance without the standard deviation due to the space constraint. More detailed results including the standard deviation can be found in Appendix C.

Table 2: The average training time of defense method against 6 backdoor attacks on $1\%$ (500 images) of clean data on CIFAR-10 training set using ResNet-18.

| Defense | FT | FP | MCR | ANP |
|---|---|---|---|---|
| Time (s) | 93.80 | 1427.1 | 286.1 | 241.5 |

The experimental results show that the proposed ANP remarkably provides almost the highest robustness against several state-of-the-art backdoor attacks. In particular, in 5 of total 6 attacks, our proposed ANP successfully reduces the ASR to lower than $1\%$ while only has a slight drop ($\sim 1.47\%$ on average) in ACC. Note that we only use $1\%$ of clean data from CIFAR-10 training set, which is extremely small. We find the standard fine-tuning (FT) with a larger learning rate ($lr = 0.02$) also provides strong robustness similar to ANP. However, it has an obvious drop in ACC since a large learning rate may destroy the features learned before and the new ones learned on limited data are poor in generalization. The poor accuracy hinders its usage in practical scenarios. In contrast, ANP has similar robustness against backdoor attacks while maintains ACC at a relatively high level at the same time. Similar to our method, Fine-pruning (FP) [24] also prunes some neurons and fine-tunes the pruned model on clean data. Specifically, It only prunes neurons in the last convolutional layer using a rule-of-thumb[6], and provides limited robustness against backdoor attacks. Differently, the proposed ANP is data-driven (*i.e.*, optimizing pruning masks based on data) and prunes neurons globally (*i.e.*, pruning neurons in all layers), leading to higher ACC and lower ASR than FP. MCR suggests mitigating backdoor by using a curve model lying on a path connection between two backdoored models in the loss landscapes. However, with limited data, it fails in providing high robustness against complex triggers (*e.g.* a random trigger sampled from Gaussian in blend backdoor attack or the dynamic trigger in input-aware attack). By contrast, ANP always reduces ASR significantly no matter what the trigger looks like. Note that ANP only prunes neurons and never changes any parameters of backdoored DNNs, while other methods all fine-tune parameters. Under this situation, ANP still provides the almost best robustness against these state-of-the-art attacks, which indicates that pure pruning (without fine-tuning) is still a promising defense against backdoor attacks.

We also compare the training time of ANP and other baselines in Table 2, which indicates the efficiency of the proposed ANP. In particular, we apply 2000 iterations for FT, FP, and ANP, and 200 epochs (*i.e.*, 800 iterations) for MCR following the open-sourced code[7]. Note that ANP is not as slow as the vanilla adversarial training (AT) since ANP only requires one extra backpropagation while PGD inside AT usually needs 10 extra backpropagations (PGD-10) in each update of model parameters. Besides, although ANP takes $2.5\times$ time compared to FT, it improves accuracy (*e.g.*, $88.98\% \rightarrow 93.44\%$ for Blend) and reduce vulnerability (*e.g.*, $10.20\% \rightarrow 0.46\%$ for Blend) significantly, making the overhead caused by ANP acceptable.

### 4.3 Ablation Studies

**Effect of Adversarial Neuron Perturbation.** Recalling Section 3.2, we apply adversarial neuron perturbations to induce the injected backdoor and then defend against backdoor attacks. Here, we evaluate the effect of the adversarial neuron perturbation in our defense strategy. In the following experiments, we defend against Badnets attack by default, unless otherwise specified. First, we optimize masks with perturbations ($\alpha = 0.2$ and $\epsilon = 0.4$, *i.e.*, ANP) and without perturbations ($\alpha = 1.0$, *i.e.*, the vanilla pruning method). Figure 2(a) shows the ACC and ASR on the test set during optimization. We find the optimization under adversarial neuron perturbations helps the backdoored model suppress ASR significantly while the vanilla one always has high ASR throughout the training process. Next, we illustrate the mask distribution after optimization. Figure 2(b) illustrates a certain fraction of masks becomes zero, which means that adversarial neuron perturbations indeed find some sensitive neurons and helps the model to remove them. In contrast, there is no mask close to $0$ for the vanilla pruning method in Figure 2(c). Finally, we prune different fractions of neurons according to their mask in Figure 2(d). For both methods, ASR becomes low after pruning more than $5\%$ of neurons[8]. However, ACC also starts to degrade significantly for the vanilla pruning method with more

---

[6]Fine-pruning prunes neurons that are less active on clean samples until there is an obvious drop in accuracy on a clean validation set.

[7]https://github.com/IBM/model-sanitization

[8]This is conceivable. As long as the backdoor-related neurons are assigned slightly smaller masks than other neurons, we still can prune them first and remove the backdoor.

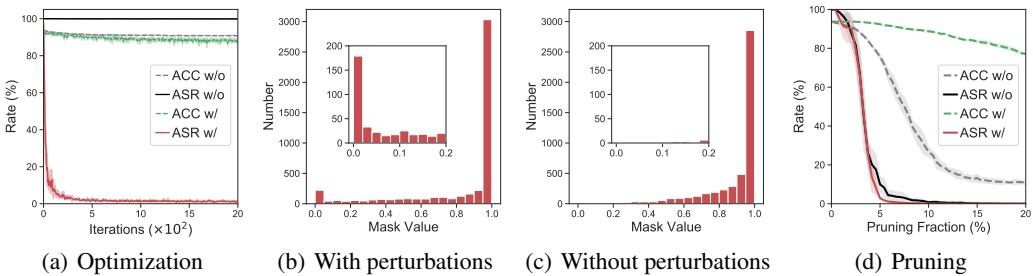

| (a) Optimization | (b) With perturbations | (c) Without perturbations | (d) Pruning |

Figure 2: Comparison between optimization with and without adversarial neuron perturbations.

Table 3: Results with small budget $\epsilon = 0.1$ against Blend attack on 500 clean images. "Neurons↓" indicates the number of neurons pruned by ANP.

| Steps | 1 | 2 | 5 | 10 |
|---|---|---|---|---|
| Time (s) | 239.9 | 359.2 | 551.8 | 943.9 |
| Neurons↓ | 159 | 188 | 235 | 259 |
| ASR (%) | 65.19 | 13.40 | 1.06 | 0.90 |
| ACC (%) | 93.62 | 93.07 | 92.95 | 92.72 |

Table 4: Results with large budget $\epsilon = 0.4$ against Blend attack on 500 clean images. "Neurons↓" indicates the number of neurons pruned by ANP.

| Steps | 1 | 2 | 5 | 10 |
|---|---|---|---|---|
| Time (s) | 241.5 | 357.2 | 557.1 | 950.1 |
| Neurons↓ | 233 | 239 | 281 | 296 |
| ASR (%) | 0.46 | 1.30 | 5.30 | 31.34 |
| ACC (%) | 93.44 | 94.07 | 93.57 | 94.28 |

than 5% of neurons pruned. Since backdoored-related neurons are still mixed with other neurons (*e.g.*, discriminant neurons), the vanilla pruning method prunes some discriminant neurons incorrectly. Meanwhile, ANP always keeps ACC at a relatively high level. In conclusion, the adversarial neuron perturbation is the key to distinguish the backdoor-related neurons from the other neurons, which thereby successfully obtains high ACC as well as low ASR.

In addition, we explore the effects of number of iterations in crafting adversarial neuron perturbations. We conduct experiments with varying numbers of steps (1/2/5/10) in ANP with a small perturbation budget ($\epsilon = 0.1$) and a large perturbation budget ($\epsilon = 0.4$) respectively. The other settings are the same as Section 4.1. The experimental results are shown in Tables 3-4. Under a small perturbation budget, with more steps for ANP, the ASR decreases with a slight drop in ACC. This is because ANP with a single step and small size is too weak to distinguish benign neurons and backdoor-related neurons, while more steps can help ANP find more backdoor-related neurons. However, under a large perturbation budget, ANP with more steps has worse robustness. This is because, with a large perturbation budget, more neurons become sensitive. As a result, ANP with more steps finds too many "suspicious" neurons, and it is unable to identify backdoor-related neurons from them. In conclusion, we can strengthen the power of ANP to find backdoor-related neurons using a larger perturbation budget or more steps. Among them, single-step ANP with a slightly larger budget is more practical due to its time efficiency, and we adopt this by default.

**Results with Varying Hyperparameters.** As mentioned in Section 3.3, the hyper-parameter $\alpha$ in ANP controls the trade-off between the accuracy on clean data and the robustness against backdoor attacks. To test the performance with different $\alpha$, we optimize masks for a Badnets ResNet-18 based on 1% of clean data using different $\alpha \in [0, 1]$ with a fixed budget $\epsilon = 0.4$. In pruning, we always prune neurons by the threshold $0.2$. As shown in the left plot of Figure 4, ANP is able to achieve a high robustness (ASR < 4%) when $\alpha \leq 0.6$. Meanwhile, ANP maintains a high natural accuracy (ACC $\geq$ 90%) as long as $\alpha \geq 0.1$. As a result, ANP behaves well with high ACC and low ASR in a range of $\alpha \in [0.1, 0.6]$.

Similarly, we also test the performance with different perturbation budgets $\epsilon$. The experiment settings are almost the same except a fixed trade-off coefficient $\alpha = 0.2$ and varying budgets $\epsilon \in [0, 1]$. For example, we can provide obvious robustness (ASR becomes 7.45%) with a small perturbation budget ($\epsilon = 0.2$) as shown in the right plot of Figure 4. However, we find the accuracy on clean data degrades with a large perturbation budget. This is because too large perturbation budget brings much difficulty to converge well and ANP only finds a poor solution, which fails in identifying these discriminant and robust neurons and prunes part of them. As a result, ACC decreases significantly. In conclusion, the

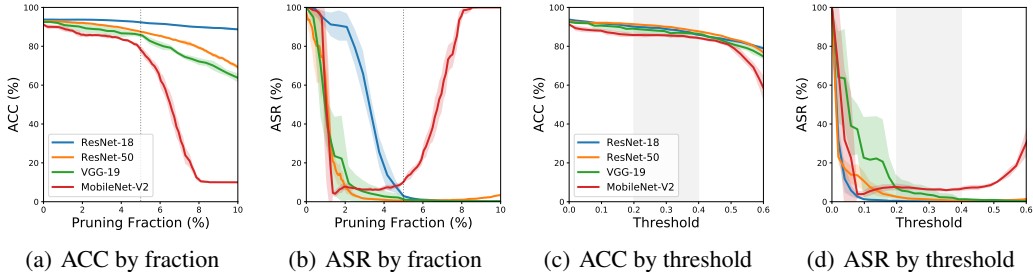

|  (a) ACC by fraction | (b) ASR by fraction | (c) ACC by threshold | (d) ASR by threshold |

Figure 3: Performance ($\pm$std over 5 random runs) of pruned models using different architectures by different pruning fractions or different thresholds.

proposed ANP is stable in a large range of the trade-off coefficient $\alpha \in [0.1, 0.6]$ and the perturbation budget $\epsilon \in [0.2, 0.7]$, demonstrating that ANP is not sensitive to hyper-parameters.

From the discussion above, we find that the hyperparameters are insensitive as shown in Figure 4 and ANP performs well across a wide range of hyperparameters, *e.g.*, trade-off coefficient $\alpha \in [0.1, 0.6]$ against Badnets attack. In addition, with varying hyperparameters, the performance trends are very consistent across different backdoor attacks as shown in Figure X in Appendix C.1. As a result, even though the attack (*e.g.*, Blend) is unknown, the defender could tune $\alpha$ against a known attack (*e.g.*, Badnets) and find the $0.2$ is a good choice. ANP with $\alpha = 0.2$ also achieves satisfactory performance against Blend attack. In conclusion, the selection of hyperparameters in ANP is not difficult.

**Pruning by Fraction or by Threshold.** Different from previous pruning methods that use a fixed pruning fraction, the proposed ANP prunes neurons by a threshold when generalized across different model architectures. To verify this, we train different Badnets models using ResNet-18 [15], ResNet-50 [15], and VGG-19 [37] with the same settings as Section 4.1. We also train a model with Badnets MobileNet-V2 with 300 epochs . We optimize the pruning masks for these models with $\epsilon = 0.4, \alpha = 0.2$ for ResNet-18, ResNet-50, and VGG-19. For MobileNet-V2, we found the perturbation budget $\epsilon = 0.4$ is too large, leading to a failure in convergence. So we apply a smaller budget $\epsilon = 0.2$ and the same trade-off coefficient $\alpha = 0.2$. We show the ACC and ASR after pruning by varying pruning fractions in Figure 3(a) and 3(b), which indicates that the suitable fraction varies across model architectures. For example, ResNet-18 only has low ASR after pruning $5\%$ of neurons, while MobileNet-V2 has a large accuracy drop on clean samples with more than $5\%$ neurons pruned. Figure 3(c) and Figure 3(d) show ACC and ASR after pruning by different thresholds respectively. When pruning by the threshold, we find there is a large overlap in $[0.2, 0.5]$ (the gray zone) in which all models have high ACC and low ASR simultaneously. That is why we adopt the strategy of pruning by the threshold in ANP. Note that, even for low-capacity models as MobileNet-V2, it is still possible to remove the injected backdoor by neuron pruning (without the fine-tuning).

**Pruning on Different Components.** We also compare the performance when applying ANP to different components inside DNNs. We prune a backdoored 4-layer CNN (2 conv layers + 1 fully-connected layer + output layer) on MNIST, and find it more efficient to prune neurons in some layers. In particular, we achieve $0.43\%$ of ASR ($99.04\%$ of ACC) when two neurons in the 2nd layer are pruned, while we only achieve $4.42\%$ of ASR ($92.48\%$ of ACC) even after pruning 150 neurons in the fully-connected layer. The experimental settings and more results can be found in Appendix C.2. This indicates that the structure of components matter in pruning-based defense. We leave this in our future work.

## 4.4 Performance under Limited Computation Resource

In practical scenarios, the defender usually has limited clean data and computation resources, bringing more difficulty to repair backdoored models. While Section 4.2 has discussed the effectiveness of the proposed ANP on the extremely small amount of data, this part focuses on the performance with limited computation resources in defense.

In particular, we optimize all masks for a Badnets ResNet-18 with varying number of iterations (20, 100, 400, and 2000) on $1\%$ of clean data (500 images) and prune the model from neurons with the small mask to neurons with the large mask. Figure 5 shows the ACC and ASR with varying pruning

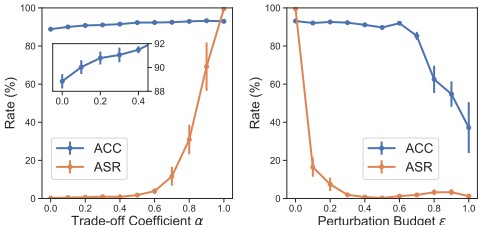
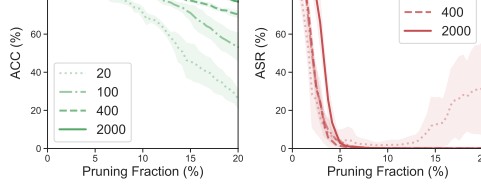

Figure 4: Performance ($\pm$ std over 5 random runs) of the pruned model by a threshold $0.2$ with different hyper-parameters (*Left*: trade-off coefficient $\alpha$; *Right*: perturbation budget $\epsilon$).

Figure 5: Performance ($\pm$std over 5 random runs) of the proposed ANP with varying pruning fractions based on different numbers of iterations.

fractions based on different numbers of iterations. We find that some backdoor-related neurons have already been distinguished just after 20 iterations. For example, after removing $4\%$ of neurons with the smallest masks, the pruned model has $6.28\%$ of ASR and $86.65\%$ of ACC. As the pruning fraction increases, ASR falls significantly first and raises again after $10\%$ of neurons are pruned. This is because ANP has not yet separated them completely due to extremely limited computation (only 20 iterations). With more than 100 iterations, this phenomenon disappears. And ANP based on 100 iterations already has comparable performance to ANP on 2000 iterations, especially when we have not pruned too many neurons ($< 8\%$).

We also find that ANP with 2000 iterations should prune more ($+1\%$) neurons to degrade ASR than ones with fewer iterations. We conjecture this is because there also exist some sensitive neurons that are unrelated to the injected backdoor[9], which are assigned with small masks at the late optimization stage. Besides, for ACC, we can see that more iterations maintain a higher ACC than fewer iterations, especially when more neurons are pruned, as the left neurons are almost the discriminant and robust ones. In conclusion, even with extremely small number of iterations (*e.g.*, 20 iterations), ANP is able to distinguish the backdoor-related neurons from other neurons and obtains satisfactory robustness against backdoor attacks.

## 5    Conclusion

In this paper, we identified a sensitivity of backdoored DNNs to adversarial neuron perturbations, which induces them to present backdoor behaviours even without the presence of the trigger patterns. Based on these findings, we proposed *Adversarial Neuron Pruning* (ANP) to prune the sensitive neurons under adversarial neuron perturbations. Comprehensive experiments show that ANP consistently removes the injected backdoor and provides the highest robustness against several state-of-the-art backdoor attacks even with a limited amount of clean data and computation resources. Finally, our work also reminds researchers that pruning (without fine-tuning) is still a promising defense against backdoor attacks.

## Broader Impact

The backdoor attack has become a threat to outsourcing training and open-sourcing models. We propose ANP to improve the robustness against these backdoor attacks, which may help to build a more secure model even trained in an untrustworthy third-party platform. Further, we do not want this paper to bring overoptimism about AI safety to the society. Since the backdoor attack is only a part of potential risks ( *e.g.*, adversarial attacks, privacy leakage, and model extraction), there is still a long way towards secure AI and trustworthy AI.

---

[9]Recalling Figure 1(b), we can see that the benign model also suffers from the adversarial neuron perturbations, although its sensitivity is much weaker than that of backdoored models.

## Acknowledgments

Yisen Wang is partially supported by the National Natural Science Foundation of China under Grant 62006153, and Project 2020BD006 supported by PKU-Baidu Fund.

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
