# A  More Implementation Details on Backdoor Attacks

In this part, we provide more implementation details on several state-of-the-art backdoor attacks. The backdoor triggers applied in our experiments are shown in Figure 6 and Figure 7.

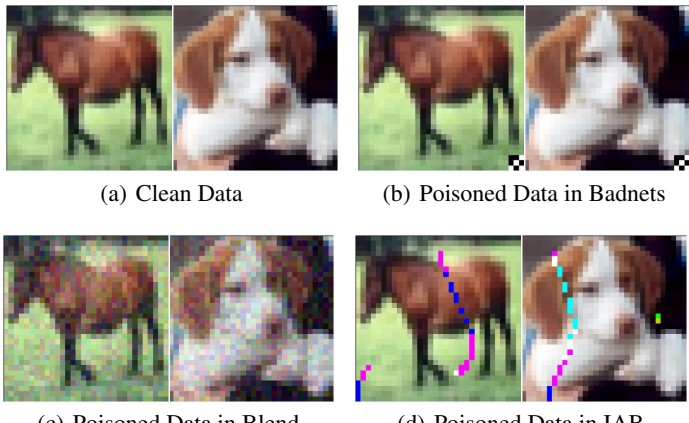

(a) Clean Data        (b) Poisoned Data in Badnets

(c) Poisoned Data in Blend        (d) Poisoned Data in IAB

Figure 6: Examples of poisoned CIFAR-10 images at the training time (*Left*) and the test time (*Right*). Given the target label (class 0, "plane"), Badnets, Blend, and IAB always attach a predefined trigger to some samples from other classes and relabel them as the target label at the training time.

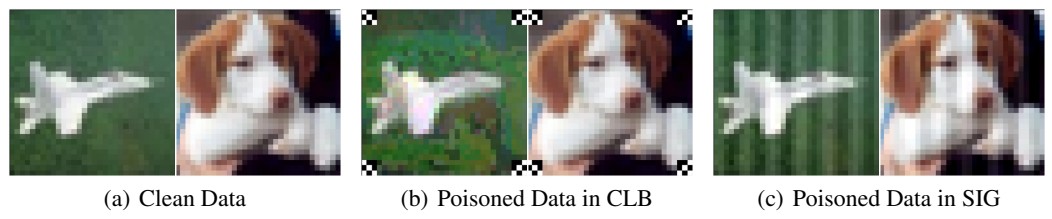

(a) Clean Data        (b) Poisoned Data in CLB        (c) Poisoned Data in SIG

Figure 7: Examples of poisoned CIFAR-10 images at the training time (*Left*) and the test time (*Right*). Given the target label (class 0, "plane"), CLB and SIG instead attach a predefined trigger to some training samples belonging to target label to avoid incorrect labels at the training time.

We always set the target label as class 0 ("plane"). The detailed implementation on several state-of-the-art backdoor attacks are as follows,

- **Badnets** [14]: The trigger is a $3 \times 3$ checkerboard at the bottom right corner of images as shown in Figure 6(b). Given the target label, we attach the trigger to $5\%$ of training samples from other classes and relabel them as the target label. After training for 200 epochs, we achieve the attack success rate (ASR) of $99.97\%$ and the natural accuracy on clean data (ACC) of $93.73\%$.

- **Blend attack** [6]: We first generate a trigger pattern where each pixel value is sampled from a uniform distribution in $[0, 255]$ as shown in Figure 6(c). Given the target class, we randomly select $5\%$ of training samples from other classes for poisoning. We attach the trigger $\mathbf{t}$ to the sample $\mathbf{x}$ using a blended injection strategy, *i.e.*, $\alpha \mathbf{t} + (1 - \alpha)\mathbf{x}$. Here, we set the blend ratio $\alpha = 0.2$. Next, we relabel them as the target label. After training for 200 epochs, we achieve ASR of $100.00\%$ and ACC of $94.82\%$.

- **Input-aware Attack (IAB)** [30]: The dynamic trigger varies across samples as shown in Figure 6(d). Based on the open-source code[10], we train the classifier and the trigger

---

[10]https://github.com/VinAIResearch/input-aware-backdoor-attack-release

generator concurrently. We apply two types of target label selection. For all-to-one, we always set the class 0 as the target label, attach the dynamic trigger to the samples from other classes and relabel them as the target label. For all-to-all, after attaching the trigger, we relabel samples from class $i$ as class $(i + 1)$[11]. Finally, we achieve ASR of $98.49\%/92.88\%$ and ACC of $93.89\%/94.10\%$ for IAB with the all-to-one target label or the all-to-all target label respectively.

- **Sinusoidal signal attack (SIG)** [1]: We superimpose a sinusoidal signal over the inputs as the trigger following Barni et al. [1]. Given the target label, we attach the trigger to $80\%$ of samples from the target class. After training for 200 epochs, we achieve ASR of $94.26\%$ and ACC of $93.64\%$.

- **Clean-label Attack (CLB)** [42]: The trigger is a $3 \times 3$ checkerboard at the four corners of images as shown in Figure 7(b). Given the target label, we attach the trigger to $80\%$ of samples from the target class. To make the backdoored DNN rely more on the trigger pattern rather than the salient features from the class, we apply adversarial perturbations to render these poisoned samples harder to classify during training following Turner et al. [42]. Specifically, we use Projected Gradient Descent (PGD) to generate adversarial perturbations with the $\ell_\infty$-norm maximum perturbation size $\epsilon = 16$ based on the open-source code[12]. After training for 200 epochs, we achieve ASR of $99.94\%$ and ACC of $93.78\%$.

## B    More Results on Adversarial Neuron Perturbations

In Section 3.2, we have demonstrated that the backdoored models are much easier to collapse and tend to predict the target label on clean samples when their neurons are adversarially perturbed. Here, we further explain that such sensitivity is from the backdoor itself rather than the unbalanced dataset caused by some attacks. For example, on CIFAR-10 training set (5000 samples for every class), Badnets relabel $5\%$ of samples from other classes. As a result, the target class has 7250 samples, while each of the other classes only has 4750 samples.

First, we create an unbalanced dataset by replacing the incorrectly-labeled samples in Badnets training set with extra samples from target class from open-source labeled data[13]. We train another benign model on the unbalanced dataset using the same training settings as the backdoored model by Badnets. Besides, we also illustrate the results of a backdoored model by SIG, which is also based on a balanced dataset since SIG only attaches the trigger to the samples from the target class.

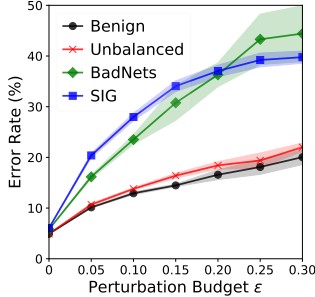

Following Section 3.2, we generate adversarial neuron perturbations for these models. Figure 8 shows the benign model based on the unbalanced dataset almost has a similar error rate compared to the original benign one, and the backdoored models by Badnets and SIG always have larger error rates with the same perturbation budget compared to the benign models. We

Figure 8: The error rate ($\pm$ std over 5 random runs) of different models under neuron perturbations.

also illustrate the proportion of different classes in predictions in Figure 9. For the backdoored model by SIG, the majority of misclassified samples have already been predicted as the target label. For the backdoored model by Badnets, much more samples are classified as the target class compared to the benign one on an unbalanced dataset. In conclusion, the injected backdoor itself results in sensitivity under adversarial neuron perturbation rather than an unbalanced dataset.

## C    More Results on Adversarial Neuron Pruning (ANP)

### C.1    Performance Trends across Different Backdoor Attacks

---

[11]For class 9, the target label is class 0.

[12]https://github.com/MadryLab/label-consistent-backdoor-code

[13]https://github.com/yaircarmon/semisup-adv

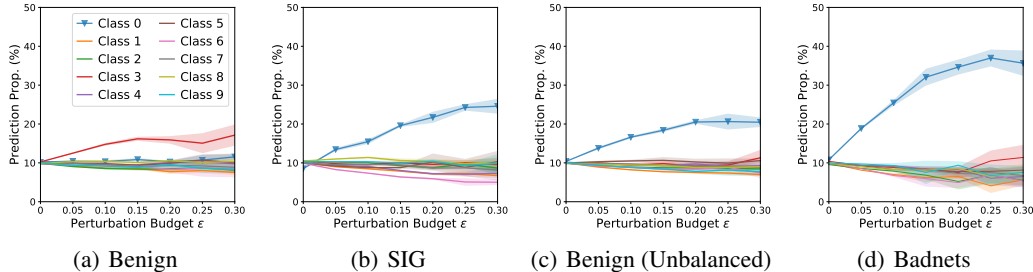

|             |             |                   |            |
| (a) Benign  | (b) SIG     | (c) Benign (Unbalanced) | (d) Badnets |

Figure 9: The proportion ($\pm$ std over 5 random runs) of different classes in predictions by a benign model, another benign one trained on an unbalanced training set, and two backdoored models. The first benign model and the SIG model are based on a balanced training set, while the second benign model and the Badnets model are based on an unbalanced training set.

In practical scenarios, the adversary may exploit any possible backdoor attacks. While the selection of hyperparameters seriously affects the effectiveness of defense, it is important to seek a selection strategy of hyperparameters for the defender. Here, we conduct experiments to illustrate the performance trends across different backdoor attacks, including Badnets, Blend, and CLB. The experimental settings are the same as Section 4.1. We evaluate the performance of ANP against three backdoor attacks with varying trade-off coefficient $\alpha$ and the results are shown in Figure 10. As $\alpha$ increases,

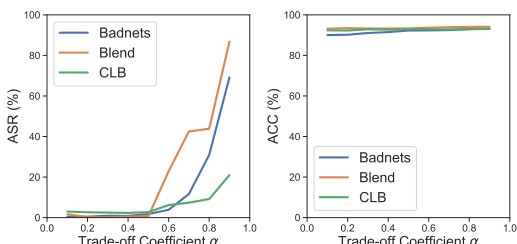

Figure 10: Performance of the pruned model by a threshold 0.2 with varying trade-off coefficient $\alpha$.

we find that ASR remains low until a common threshold ($\alpha = 0.5$) is exceeded. Besides, ACC always keeps high with different $\alpha$. Thus, with varying hyperparameters, the performance trends are very consistent across different backdoor attacks. Besides, we also find the hyperparameters are insensitive and ANP performs well across a wide range of hyperparameters in Figure 4 of Section 4.3.

Therefore, for simplicity, we always set the same hyperparameters (*e.g.*, $\alpha = 0.2$ as shown in Section 4.1) in this paper against various backdoor attacks unless otherwise specified. In practical scenarios, the defender could tune the hyperparameters in a straightforward way: assuming the unknown attack is Blend attack, the defender could tune against a known attack (*e.g.*, Badnets) and find the 0.2 is a good choice. ANP with also achieves satisfactory performance against Blend attack.

### C.2    Pruning on Different Components

In this part, we explore the performance when applying ANP to different components inside DNNs. We conduct a toy experiment based on a 5-layer fully-connected (FC) neural network to show the effectiveness of the proposed ANP. We train the Badnets model on a poisoned MNIST training set whose backdoor trigger is a square at the bottom right and the injection rate is $10\%$. After training, the ACC is $98.08\%$, and the ASR is $100.00\%$. We apply ANP ($\alpha = 0.5, \epsilon = 0.3$) to the whole model, we obtain a repaired model with $93.86\%$ of ACC and $1.93\%$ of ASR after pruning 560 neurons (2000 in total).

Table 5: Results on MNIST based on a 5-layer FC neural network. "Neurons↓" indicates the number of neurons pruned by ANP, and "Total" indicates the number of all neurons.

|           | Before | ANP  |
| --------- | ------ | ---- |
| Neurons↓  | 0      | 560  |
| Total     | 2000   | 2000 |
| ASR (%)   | 100.00 | 1.93 |
| ACC (%)   | 98.08  | 93.86 |

Table 6: Results on MNIST of different components based on a 4-layer CNN. "Neurons↓" indicates the number of neurons pruned by ANP, and "Total" indicates the number of all neurons.

| | Before | ANP on all | ANP on Conv | ANP on FC |
|---|---|---|---|---|
| Neurons↓ | 0 | 8 | 2 | 150 |
| Total | 560 | 560 | 48 | 512 |
| ASR (%) | 99.99 | 0.32 | 0.43 | 4.42 |
| Acc (%) | 99.34 | 99.22 | 99.04 | 92.48 |

Next, we conduct a similar experiment on the same poisoned MNIST training set based on a 4-layer CNN (2 conv layers + 1 FC layer + output layer) to compare the performance between the conv layers and the FC layer. After training, the backdoored model has 99.34% of ACC and 99.99% of ASR. We apply ANP on conv layers and the FC layer respectively. In particular, If we only apply ANP on conv layers, we achieve 99.04% of ACC and 0.43% of ASR just after pruning 2 neurons in the second conv layer. If only applying ANP on the FC layer, we achieve 92.48% of ACC and 4.42% of ASR even after pruning 150 neurons (512 in total). We conjecture this is because, for a simple trigger pattern, the neurons most related to the backdoor tend to locate in the shallow layers (*i.e.*, conv layers), which makes pruning on the conv layers has better performance. Therefore, the performance of a component is dependent on its structure.

## C.3   Benchmarking Results Based on Varying Fraction of Clean Data

In this part, we provide more experimental results based on varying fraction of CIFAR-10 clean data, including $10\%$ (5000 images), $1\%$ (500 images), and $0.1\%$ (50 images), in Tables 7-9.

We find all baselines suffer from difficulty in defense with a limited amount of data, *i.e.*, ASR significantly increases when the number of clean samples decreases. However, the proposed ANP always degrades ASR lower than $5\%$ even with $0.1\%$ of clean data (50 images), which indicates that ANP provides satisfactory robustness against backdoor attacks. Besides, when we have enough amount of data (*e.g.*, $10\%$ of clean data), the ACC drop in ANP is negligible ($< 1\%$).

Note that Fine-tuning (FT) and Fine-pruning (FP) have better natural accuracy (ACC) based on $1\%$ of clean data than $10\%$ of clean data. We conjecture this is because it is much easy to overfit to a small number of samples (*e.g.*, 500 images) and keep the original natural ACC better. However, we always achieve higher ASR based on less amount of data.

Table 7: Performance ($\pm$ std over 5 random runs) of 4 defense methods against 6 backdoor attacks on $10\%$ (5000 images) of clean data on CIFAR-10 training set using ResNet-18.

| Metric | Defense | Badnets | Blend | IAB-one | IAB-all | CLB | SIG |
|---|---|---|---|---|---|---|---|
| ACC | Before | 93.73 | 94.82 | 93.89 | 94.10 | 93.78 | 93.64 |
| | FT (0.01) | 87.74±1.14 | 88.39±0.96 | 85.59±1.11 | 86.54±0.57 | 87.12±1.53 | 86.95±1.12 |
| | FT (0.02) | 85.56±0.89 | 86.25±0.94 | 83.75±1.15 | 85.62±1.59 | 84.70±0.88 | 85.80±1.36 |
| | FP | 86.95±1.03 | 88.26±0.57 | 85.64±0.97 | 85.26±2.53 | 86.89±1.21 | 87.47±1.41 |
| | MCR(0.3) | 89.17±0.33 | 89.45±0.53 | 87.01±0.10 | 88.53±1.12 | 89.78±0.21 | 85.88±0.56 |
| | ANP | **93.01±0.22** | **93.86±2.13** | **93.40±0.11** | **93.56±0.09** | **93.37±0.20** | **93.65±0.17** |
| ASR | Before | 99.97 | 100.0 | 98.49 | 92.88 | 99.94 | 94.26 |
| | FT (0.01) | 2.60±0.98 | 0.86±1.29 | 4.74±1.74 | 1.91±0.26 | 1.56±0.19 | 0.86±1.29 |
| | FT (0.02) | 2.13±1.23 | **0.27±0.22** | 2.67±0.54 | 2.15±0.21 | 2.27±0.75 | 0.92±0.29 |
| | FP | **1.83±0.42** | 0.38±0.43 | 1.97±0.42 | 6.60±1.99 | **1.97±0.42** | 0.62±0.23 |
| | MCR(0.3) | 5.70±1.21 | 10.57±2.21 | 15.23±1.43 | 17.17±2.71 | 3.77±1.12 | 0.52±0.21 |
| | ANP | 3.34±1.43 | 2.13±1.47 | **1.32±0.29** | **0.79±0.04** | 3.65±1.48 | **0.28±0.14** |

Table 8: Performance ($\pm$ std over 5 random runs) of 4 defense methods against 6 backdoor attacks on 1% (500 images) of clean data on CIFAR-10 training set using ResNet-18.

| Metric | Defense | Badnets | Blend | IAB-one | IAB-all | CLB | SIG |
|---|---|---|---|---|---|---|---|
| ACC | Before | 93.73 | 94.82 | 93.89 | 94.10 | 93.78 | 93.64 |
| | FT (0.01) | 90.48±0.36 | 92.12±0.33 | 88.68±0.26 | 89.06±0.22 | 91.26±0.24 | 91.19±0.12 |
| | FT (0.02) | 87.23±0.51 | 88.98±0.32 | 84.85±0.57 | 83.77±0.32 | 88.25±0.20 | 88.63±0.33 |
| | FP | **92.18±0.09** | 92.40±0.04 | 91.57±0.12 | 92.28±0.05 | 91.91±0.12 | 91.64±0.12 |
| | MCR(0.3) | 85.95±0.21 | 88.26±0.35 | 86.30±0.10 | 84.53±1.12 | 86.87±0.33 | 85.88±0.56 |
| | ANP | 90.20±0.41 | **93.44±0.43** | **92.62±0.32** | **92.79±0.16** | **92.67±0.16** | **93.40±0.12** |
| ASR | Before | 99.97 | 100.0 | 98.49 | 92.88 | 99.94 | 94.26 |
| | FT (0.01) | 11.70±4.10 | 47.17±19.51 | 0.99±0.55 | 1.36±0.31 | 12.51±8.16 | 0.40±0.23 |
| | FT (0.02) | 2.95±0.72 | 10.20±20.64 | 1.70±0.19 | 1.83±0.38 | **1.17±0.10** | 0.39±0.04 |
| | FP | 5.34±1.94 | 65.39±14.55 | 20.73±10.35 | 32.36±1.92 | 3.40±0.69 | 0.32±0.21 |
| | MCR(0.3) | 5.70±2.31 | 13.57±1.29 | 30.23±3.47 | 35.17±3.76 | 12.77±2.12 | 0.52±0.10 |
| | ANP | **0.45±0.17** | **0.46±0.23** | **0.88±0.14** | **0.86±0.05** | 3.98±2.83 | **0.28±0.25** |

Table 9: Performance ($\pm$ std over 5 random runs) of 4 defense methods against 6 backdoor attacks on 0.1% (50 images) of clean data on CIFAR-10 training set using ResNet-18.

| Metric | Defense | Badnets | Blend | IAB-one | IAB-all | CLB | SIG |
|---|---|---|---|---|---|---|---|
| ACC | Before | 93.73 | 94.82 | 93.89 | 94.10 | 93.78 | 93.64 |
| | FT (0.01) | 88.90±1.90 | 91.21±0.74 | 88.17±2.15 | 84.21±2.89 | **91.10±0.40** | **90.00±0.79** |
| | FT (0.02) | 82.06±2.84 | 86.79±6.81 | 75.28±4.92 | 60.45±9.36 | 88.11±1.76 | 84.37±2.94 |
| | FP | **90.69±0.23** | **91.29±0.17** | **90.74±0.23** | **90.26±0.24** | 89.27±0.66 | 89.96±0.90 |
| | MCR(0.3) | 80.21±5.19 | 81.35±3.19 | 79.10±3.15 | 82.53±1.12 | 77.33±4.33 | 85.88±0.56 |
| | ANP | 86.62±0.64 | 88.12±0.38 | 88.54±1.25 | 85.66±1.27 | 86.73±0.99 | 89.54±0.28 |
| ASR | Before | 99.97 | 100.0 | 98.49 | 92.88 | 99.94 | 94.26 |
| | FT (0.01) | 82.58±24.42 | 100.00±0.00 | 1.24±0.53 | 14.69±22.87 | 50.25±40.73 | 12.59±11.57 |
| | FT (0.02) | 16.27±12.55 | 99.99±0.03 | 2.29±0.85 | 3.78±1.01 | 22.71±43.19 | 0.07±0.07 |
| | FP | 30.71±16.72 | 99.95±0.02 | 30.82±51.57 | 42.38±2.25 | 1.35±1.00 | **0.13±0.03** |
| | MCR(0.3) | 33.70±3.39 | 43.57±1.29 | 52.23±3.67 | 53.17±5.21 | 12.77±3.77 | 1.10±0.01 |
| | ANP | **0.45±0.17** | **5.70±8.87** | **1.84±0.76** | **1.43±0.28** | **0.23±0.10** | 2.85±1.19 |