# OpenReview forum: "Adversarial Neuron Pruning Purifies Backdoored Deep Models"
_NeurIPS.cc/2021/Conference — NeurIPS 2021 Poster_

### Official Review · Reviewer_jkbL · 2021-07-07

**Rating:** 6
**Confidence:** 5

**Summary:**

This paper proposed Adversarial Neuron Pruning (ANP) to detect the backdoored DNNs. APN prunes the most sensitive neurons under adversarial neuron perturbations. Extensive experiments show that ANP provides state-of-the-art defense performance.


**Limitations And Societal Impact:**

My main concerns:
1. On page 3, you take a fully-connected network as an example to show the training procedure. However, in my opinion, applying ANP on fully-connected layer and convolutional layer are different. FC layer has more neurons while conv layer has much fewer neurons. Thus, compared to conv layer, I think the FC layer can have a better performance but hard to train. Can you conduct a small experiment to show the results of ANP on FC layers?
2. AT is very show, can you report the training time of your methods and other baselines?
3. In each round, you apply PGD only once (PGD-1), how is the performance if you use PGD-10[1] (i.e., apply PGD 10 times each round) instead?
[1] Madry A, Makelov A, Schmidt L, et al. Towards deep learning models resistant to adversarial attacks[J]. arXiv preprint arXiv:1706.06083, 2017.


**Main Review:**

This work is inspired by adversarial training (AT), which tries to find the most adversarial data around the original data.
Different from adversarial training, this work finds the most adversarial training parameters.
I like the idea of utilizing AT techniques in backdoor attacks, which can better enhance the defense in backdoor attacks.


**Time Spent Reviewing:**

3

---

> ### Author Response · Authors · 2021-08-10
> **Response to Reviewer jkbL**
>
> **Q1**: Can you conduct a small experiment to show the results of ANP on FC layers?
>
> **A1**: Following your suggestions, we conduct a toy experiment based on a 5-layer fully-connected (FC) neural network to show the effectiveness of the proposed ANP. We train the Badnets model on a poisoned MNIST training set whose backdoor trigger is the square at the bottom right and the injection rate is 10%. After training, the ACC is 98.08%,and the ASR is 100.00%. We apply ANP ($\alpha=0.5, \epsilon=0.3$) to the whole model, we obtain a repaired model with 93.86% of ACC and 1.93% of ASR after pruning 560 neurons (2000 in total).
>
> **Table: The performance on MNIST based on a 5-layer FC neural network**
>
> | | Before | ANP |
> | :----------- | :--------- | :---------- |
> | #Pruned neuron (#total neuron) | 0 (2000) | 560 (2000) |
> | ASR (%) | 100.00 | 1.93 |
> | ACC (%) | 98.08  | 93.86 |
>
> Next, we conduct a similar experiment on the same poisoned MNIST training set based on a 4-layer CNN (2 conv layer + 1 FC layer + output layer) to compare the performance between the conv layers and the FC layer. After training, the backdoored model has 99.34% of ACC and 99.99% of ASR. We apply ANP on conv layers and the FC layer respectively. In particular, If we only apply ANP on conv layers, we achieve 99.04% of ACC and 0.43% of ASR just after pruning 2 neurons in the second conv layer. If only applying ANP on the FC layer, we only achieve 92.48% of ACC and 4.42% of ASR even after pruning 150 neurons (512 in total). We conjecture this is because, for a simple trigger pattern, the neurons most related to the backdoor tend to locate in the shallow layers (i.e., conv layers), which makes the conv layers have better performance. Therefore, the performance of a layer is more dependent on the distribution of the backdoor-related neurons in the neural network rather than its structure.
>
> **Table: The performance on MNIST of different layers based on a 4-layer CNN**
>
> | | Before | ANP on all | ANP on conv | ANP on FC |
> | :----------- | :--------- | :---------- | :----------- | :--------- |
> | # Pruned neuron ( #total neuron) | 0 (560) | 8 (560) | 2 (48) | 150 (512) |
> | ASR (%) | 99.99 | 0.32 | 0.43 | 4.42 |
> | ACC (%) | 99.34 | 99.22 | 99.04 | 92.48 |
>
> ***
>
> **Q2**: Can you report the training time of ANP and other baseliness?
>
> **A2**: The training time of ANP and other baselines are reported in the table below, where we apply 2000 iterations for fine-tuning (FT), fine-pruning (FP), and the proposed ANP, and 200 epochs (i.e., 800 iterations) for mode connectivity repair (MCR) following the open-sourced code (https://github.com/IBM/model-sanitization).
>
> Note that ANP is not as slow as the vanilla adversarial training (AT) since ANP only requires one extra backpropagation while PGD inside AT usually needs 10 extra backpropagations in each update of model parameters. Besides, although ANP takes 2.5$\times$ time compared to FT, it improves accuracy and robustness significantly, making the overhead caused by ANP acceptable.
>
> **Table: The training time and performance of different defenses against Blend attack on 500 images**
>
> | | FT | FP | MCR | ANP |
> | :----------- | :--------- | :---------- | :----------- | :--------- |
> | Time (s) | 93.8 | 1427.1 | 286.1 | 241.5 |
> | ASR (%) | 47.17 | 65.39 | 13.57 | 0.46 |
> | ACC (%) | 92.12 | 92.40 | 88.26 | 93.44 |
>
> ***
>
> **Q3**: How is the performance if we use PGD-10?
>
> **A3**: We conduct additional experiments with varying numbers of steps (1/2/5/10) in ANP with a small perturbation budget ($\epsilon=0.1$) and a large perturbation budget ($\epsilon=0.4$) respectively. The other settings are the same as Section 4.1. The experimental results are shown in the tables below.
>
> Under a small perturbation budget, with more steps for ANP, the ASR decreases with a slight drop in ACC. This is because ANP with a single step and small size is too weak to distinguish benign neurons and backdoor-related neurons, while more steps can help ANP find more backdoor-related neurons.
>
> However, under a large perturbation budget, ANP with more steps has worse robustness. This is because, with a large perturbation budget, more neurons become sensitive. As a result, ANP with more steps finds too many “suspicious” neurons, and it is unable to identify backdoor-related neurons from them.
>
> In conclusion, we can strengthen the power of ANP to find backdoor-related neurons using a larger perturbation budget or more steps. Among them, single-step ANP with a slightly larger budget is more practical due to its time efficiency.
>
> **Table: Results with small budget ($\epsilon=0.1$) against Blend attack on 500 clean images**
>
> | # Steps | 1 | 2 | 5 | 10 |
> | :----------- | :--------- | :---------- | :----------- | :--------- |
> | Time | 239.9 | 359.2 | 551.8 | 941.9 |
> | # Pruned neurons | 159 | 188 | 235 | 259 |
> | ASR | 65.19 | 13.40 | 1.06 | 0.90 |
> | ACC | 93.62 | 93.07 | 92.95 | 92.72 |
>
> **Table: Results with large budget ($\epsilon=0.4$) against Blend attack on 500 clean images**
>
> | # Steps | 1 | 2 | 5 | 10 |
> | :----------- | :--------- | :---------- | :----------- | :--------- |
> | Time | 241.5 | 357.2 | 557.1 | 950.1 |
> | # Pruned neurons | 233 | 239 | 281 | 296 |
> | ASR | 0.46 | 1.30 | 5.30 | 31.34 |
> | ACC | 93.44 | 94.07 | 93.57 | 94.28 |

---

### Official Review · Reviewer_H7Ya · 2021-07-12

**Rating:** 7
**Confidence:** 3

**Summary:**

This paper proposes a simple yet effective method to purify the injected backdoor based on a small portion of clean data without access to training phase.
The proposed method dubbed Adversarial Neuron Pruning (ANP) prunes some sensitive neurons to defend backdoor attacks. Empirical results validate ANP can significantly improve robustness against various backdoor attacks.

**Ethical Concerns:**

No ethical issues.

**Limitations And Societal Impact:**

Limitations: Please carefully refer to "Main Review".
This paper targets to defend backdoor attacks. Therefore, it has no negative societal impacts.

**Main Review:**

Pros:
1. The proposed method seems to be well motivated. Studying the relation between the backdoor trigger patten and the sensitivity of neurons under adversarial neuron perturbations is novel.
2. The experimental results comprehensively show that ANP seems to achieve the state-of-the-art robustness against several strong backdoor attacks. In addition, ANP can be conveniently implemented to repair the model without fine-tuning.

Cons:
1. It seems hard to intuitively understand why the sensitivity of neurons under adversarial neuron perturbation becomes closely related to the injected backdoor although this phenomenon is empirically validated.
2. Does the neuron pruning incur the degradation of generalization? It is better to compare and report the utility of purified model as well.

**Time Spent Reviewing:**

3

---

> ### Author Response · Authors · 2021-08-10
> **Response to Reviewer H7Ya**
>
> **Q1**: How to intuitively understand why the sensitivity of neurons under adversarial neuron perturbation becomes closely related to the injected backdoor?
>
> **A1**: Here, we provide some intuitive understanding:
>
> Commonly, backdoor triggers create “shortcuts” from clean samples into the region belonging to the target label as stated in Wang et. al [1]. With the presence of the trigger, the backdoor-related neuron should make a large contribution to overwhelm other discriminative neurons, so as to predict the target label.
>
> While under neuron perturbations, all neurons might be active again. Once the backdoor-related neurons are active again (we give an intuitive explanation in Lines 145-150), they can easily have a larger change to the output compared to other discriminative neurons. As a result, they probably contribute more sensitivity under adversarial neuron perturbations.
>
> *[1] Bolun Wang, Yuanshun Yao, Shawn Shan, Huiying Li, Bimal Viswanath, Haitao Zheng, and Ben Y Zhao. Neural cleanse: Identifying and mitigating backdoor attacks in neural networks. In S&P, 2019.*
>
> ***
> **Q2**: Does the neuron pruning incur the degradation of generalization?
>
> **A2**: Actually, it slightly hurts the generalization of the purified model. As shown in Table 1, the degradation of natural accuracy (ACC) on the clean test set is 0.24% - 3.53% (1.47% on average).  Meanwhile, we successfully degrade attack success rate (ASR) by 92.02% - 99.54% (96.44% on average). In some safety-critical applications, this is acceptable.

---

> > ### Comment · Reviewer_H7Ya · 2021-08-31
> > **Thanks for your response**
> >
> > Thank you for the detailed answers. The answers to Q1 and Q2 are satisfactory. I agree with your points and appreciate the technique of ANP drastically degrades ASR. Therefore, I am willing to keep my score after rebuttal.

---

### Official Review · Reviewer_BQVU · 2021-07-14

**Rating:** 7
**Confidence:** 5

**Summary:**

The paper focuses on the problem of backdoor defense without any knowledge of triggers, which is an emerging topic recently (models are trained at the third party platform). The authors proposed to use adversarially neuron perturbation to find the most sensitive neuron to backdoor trigger, then used Adversarial Neuron Pruning (ANP) to purify the model. The experiments are complete and comprehensive to validate the effectiveness of ANP.


**Limitations And Societal Impact:**

No flaw found

**Main Review:**

Strength:
- It is interesting to know how to trigger the backdoor effect without knowing any knowledge of the trigger. The authors found that through adversarially perturbing neurons, backdoored DNNs can present backdoor behaviors and is much easier to output misclassification than normal DNNs. The methods are novel and the findings are inspiring.
- The proposed Adversarial Neuron Pruning (ANP) is simple but effective. ANP prunes the most sensitive neurons under adversarial neuron perturbations without fine-tuning. This work also convinces the community that pruning (without fine-tuning) is still a promising defense against backdoor attacks.
- The experiments are comprehensive. Even given a limited amount of clean data and computation resources, ANP consistently removes the injected backdoor and provides the highest robustness against several state-of-the-art backdoor attacks.

Weakness:
- How to choose the hyperparameters like alpha?
- What is the overhead of ANP?
- Can ANP be extended to the defense during training? E.g., the sensitive neurons will be pruned along with the training process.

**Time Spent Reviewing:**

30 minutes

---

> ### Author Response · Authors · 2021-08-10
> **Response to Reviewer BQVU**
>
> **Q1**: How to choose the hyperparameters like alpha?
>
> **A1**: The selection of hyperparameters in ANP is simple because:
> 1. the hyperparameters are insensitive as shown in Figure 3 and ANP performs well across a wide range of hyperparameters, e.g., trade-off coefficient $\alpha \in [0.1, 0.6]$ against Badnets attack.
> 2. With varying hyperparameters, the performance trends are very consistent across different backdoor attacks. We evaluate the performance of ANP against three backdoor attacks with varying trade-off coefficient and the results are shown in the tables below. As $\alpha$ increases, we find that ASR remains low until a common threshold ($\alpha=0.5$) is exceeded. Besides, ACC always keeps high with different $\alpha$.
>
> **Table: Attack success rate (ASR) with different $\alpha$**
>
> | Alpha | 0.1 | 0.2 | 0.3 | 0.4 | 0.5 | 0.6 | 0.7 | 0.8 | 0.9 |
> |:--------|:-----|:-----|:-----|:-----|:-----|:-----|:-----|:-----|:-----|
> | Badnets |   0.45 |   0.45 |   0.92 |   0.84 |   1.77 |   3.85 |  11.63 | 30.92 | 69.20 |
> | Blend     |   1.65 |   0.25 |   0.18 |   0.49 |   0.71 |  22.83|  42.50 | 43.84 | 86.78 |
> | CLB       |   2.99 |   2.65 |   2.48 |   2.33 |   2.72 |   6.18 |   7.36  |  9.15  | 20.96 |
>
> **Table: Accuracy (ACC) with different different $\alpha$**
>
> | Alpha | 0.1 | 0.2 | 0.3 | 0.4 | 0.5 | 0.6 | 0.7 | 0.8 | 0.9 |
> |:--------|:-----|:-----|:-----|:-----|:-----|:-----|:-----|:-----|:-----|
> | Badnets |  90.03 | 90.20 | 91.05 | 91.49 | 92.28 | 92.34 | 92.48 | 92.94 | 93.24 |
> | Blend     |  93.18 | 93.44 | 93.28 | 93.26 | 93.30 | 93.70 | 93.92 | 94.04 | 94.14 |
> | CLB       |  92.37 | 92.28 | 92.87 | 92.69 | 92.95 | 92.96 | 92.81 | 92.99 | 93.06 |
>
> Therefore, for simplicity, we always set the same hyperparameters (e.g., $\alpha=0.2$ as shown in Section 4.1) in this paper against various backdoor attacks unless otherwise specified.
>
> In practical scenarios where the adversary may exploit any possible backdoor attacks, the defender could tune the hyperparameters in a straightforward way: assuming the unknown attack is Blend attack, the defender could tune $\alpha$ against a known attack (e.g. Badnets) and find the 0.2 is a good choice. ANP with $\alpha=0.2$ also achieves satisfactory performance against Blend attack.
>
> ***
>
> **Q2**: What is the overhead of ANP?
>
> **A2**: The training time of ANP and other baselines are reported in the table below, where we apply 2000 iterations for fine-tuning (FT), fine-pruning (FP), and the proposed ANP, and 200 epochs (i.e., 800 iterations) for mode connectivity repair (MCR) following the open-sourced code (https://github.com/IBM/model-sanitization).
>
> Note that ANP is not as slow as the vanilla adversarial training (AT) since ANP only requires one extra backpropagation while PGD inside AT usually needs 10 extra backpropagations in each update of model parameters. Besides, although ANP takes 2.5$\times$ time compared to FT, it improves accuracy and robustness significantly, making the overhead caused by ANP acceptable.
>
> **Table: The training time and performance of different defenses against Blend attack on 500 images**
>
> | | FT | FP | MCR | ANP |
> |:-----|:-----|:-----|:-----|:-----|
> |Time (s) | 93.8 | 1427.1 | 286.1 | 241.5 |
> | ASR (%) | 47.17 | 65.39 | 13.57 | 0.46 |
> | ACC (%) | 92.12 | 92.40 | 88.26 |93.44|
>
> ***
>
> **Q3**: Can ANP be extended to the defense during training?
>
> **A3**: It is likely to be. We could prune suspicious neurons on a clean val set and learn weight parameters on the training set alternatively. During training, once the model is injected with a backdoor by the poisoned data, ANP could prune the backdoor-related neurons to repair the model.
>
> However, as long as these poisoned data exist on the training set, the repaired model might learn the backdoor pattern again, invalidating the previous pruning. We will continue to explore this direction in the future.

---

### Official Review · Reviewer_5ThB · 2021-07-20

**Rating:** 6
**Confidence:** 3

**Summary:**

This paper propose to distinguish and prune neurons which are sensitive to adversarial perturbations so that backdoor attacks can be better defended. Different from traditional methods which focuses on the perturbations on the input images, this paper explores the correlations between the sensitive neurons to injected backdoor and the multiplicative perturbation on neurons, and proposes Adversarial Neuron Pruning (ANP) algorithm to prune the these neurons to improve the robustness against backdoor attacks.

**Limitations And Societal Impact:**

This work could benefit AI safety in terms of outsourcing training and open-sourcing models.

**Main Review:**

Overall, I think this paper is clear and well-organized. Proposed algorithm seems natural. Extensive experiments are conducted to demonstrate the effectiveness of proposed algorithm. The comparison with other baselines show superiority. My only concern lies in the discussion in Section 3.2.

The authors argue that there exist a strong connection between the neurons which are sensitive to adversarial neuron perturbations and those to injected backdoor, which is kind of counter intuitive since adversarial attacks, such as PGD, always generate perturbations with different patterns for different samples, and could activate different neurons. There is no sufficient analysis or empirical observations to support this assumption.


**Time Spent Reviewing:**

4

---

> ### Author Response · Authors · 2021-08-10
> **Response to Reviewer 5ThB**
>
> **Q1**: There is no sufficient analysis or empirical observations to support the connection between the neurons which are sensitive to adversarial neuron perturbations and those to injected backdoor.
>
> **A1**:  There exist some neurons responsible for backdoor as found by previous work [1, 2]. These backdoor-related neurons are probably shared by different poisoned inputs because the trigger is usually fixed across many examples. Meanwhile, PGD-based adversarial perturbations are sample-wise and likely activate different neurons when using different adversarial inputs. This is the key difference of these two scenarios (backdoor vs. PGD).
>
> For the backdoor scenario, the adversarial neuron perturbation is calculated on the whole val set in Section 3.2, which is also shared across different inputs. Intuitively, these backdoor-related neurons are probably most sensitive under such a shared neuron perturbation across different inputs.
>
> For empirical evidence, Figure 1 illustrates that backdoored models (i.e., models with backdoor-related neurons) are more sensitive under adversarial neuron perturbations (higher misclassification rate), and they tend to misclassify clean inputs as the injected target label (the backdoor behaviour), which indicates the connection of neurons that are sensitive to adversarial neuron perturbations and the injected backdoor.
>
> *[1] Tianyu Gu, Brendan Dolan-Gavitt, Siddharth Garg. BadNets: Identifying Vulnerabilities in the Machine Learning Model Supply Chain. arXiv Preprint arXiv:1708.06733, 2017.*
>
> *[2] Kang Liu, Brendan Dolan-Gavitt, Siddharth Garg. Fine-Pruning: Defending Against Backdooring Attacks on Deep Neural Networks. In RAID, 2018.*

---

> > ### Comment · Reviewer_5ThB · 2021-08-31
> > **Response to rebuttal**
> >
> > Thanks for the detailed response. I keep my score after rebuttal.

---

### Decision · Program_Chairs · 2021-09-27

**Decision:**

Accept (Poster)

**Comment:**

This paper focuses on the problem of backdoor defense without any knowledge of triggers. The proposal is an adversarially neuron perturbation to find the most sensitive neuron to backdoor trigger, and then use Adversarial Neuron Pruning (ANP) to purify the model. The philosophy behind sounds quite interesting to me, namely, exploring the correlations between the sensitive neurons to injected backdoor and the multiplicative perturbation on neurons. This philosophy leads to a novel algorithm design I have never seen.

The clarity and novelty are above the bar of NeurIPS. While the reviewers had some concerns on the significance, the authors did a particularly good job in their rebuttal. Thus, all of us have agreed to accept this paper for publication! Please include the additional experimental results in the next version.